# Exploring the Applications of Neural Cellular Automata in Molecular Sciences

**Sebastian Pagel**
Department of Chemistry
University of Glasgow
Glasgow, 11 Chapel Lane
s.pagel.1@research.gla.ac.uk

**Leroy Cronin**
Department of Chemistry
University of Glasgow
Glasgow, 11 Chapel Lane
lee.cronin@glasgow.ac.uk

## Abstract

In recent years, Cellular Automata have been merged with developments in deep learning to replace the traditional update rules with a neural network. These Neural Cellular Automata (NCAs) have been applied for 2D, and 3D object generation, morphogenesis, as well as the orchestration of goal-directed behavioural responses. While there have been numerous examples of applying NCAs to emoji-like, and common gameplay objects (like houses or trees in Minecraft), their adaption to molecule representations has yet to be explored. In this work, we present an adaptation of 3D NCAs to voxelized representations of small- and bio-molecules. We present three exemplary applications of NCAs to design small-molecule interactors, reconstruct missing parts of protein backbones, and model physical transformations.

## 1 Introduction

Interactions between small-molecules, biomolecules, cells or even entire organisms govern, and determine the development of every life form. The shape and chemical composition of a protein determines its interactions with other biomolecules, modulators, or drugs. Light-molecule interactions can be used to control the state of some molecules or activate them for photochemical reactions. Recently, Neural Cellular Automata (NCA) have been developed as an extension to the classical Cellular Automata to generate complex two- and three-dimensional structures. Through the exchange of predefined update rules, as used in Conway's Game of Life [4], with a Deep Neural Network, these NCAs are able to learn the local interaction rules to generate abstract structures as found in Minecraft or commonly used emojis in human interactions [9, 17]. They have also been extended to guide dynamic, goal-guided behaviour of 2-dimensional objects as in [18]. In this work, we present an adaptation of these ideas to voxelized molecule representations. We show that NCAs can be conditioned on protein pockets to generate small-molecule interactors and model physical transformations. We also show that other than in previous works, some of the learned rules can be generalized to reconstruct a diverse set of 3D structures without the need to find a new set of update rules. Examples of the application of NCAs are presented on voxelized electron densities decorated with electrostatic potentials, as well as multi-class atom channel representations of a diverse set of molecules. The primary aim of this work is to underscore the contributions and adaptability of techniques recently developed in the field of NCAs within the context of molecular sciences, without making claims of superiority over traditional or other Deep Learning methods. The examination of limitations and comparisons is deferred for future exploration.

37th Conference on Neural Information Processing Systems (NeurIPS 2023).

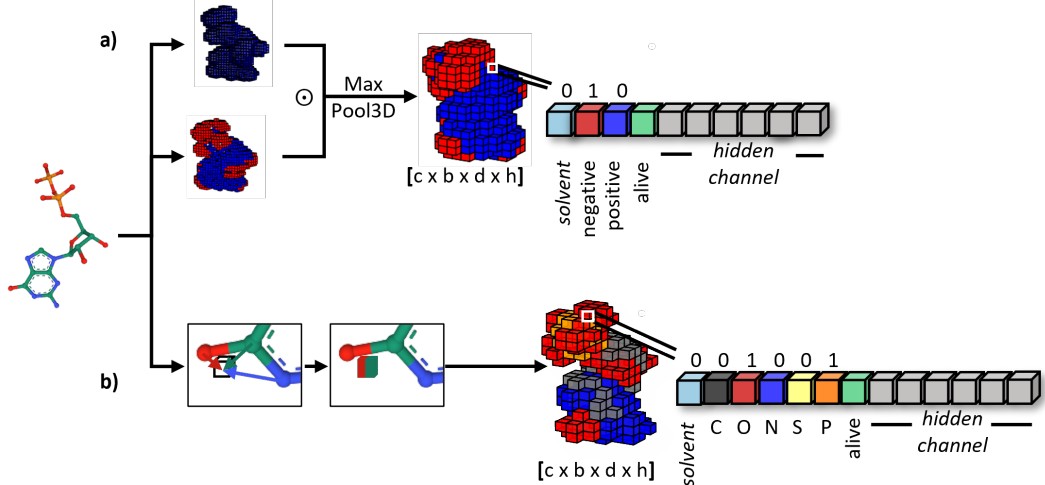

Figure 1: Voxelization of molecules. Molecules (shown here is GDP extracted from PDB: 1d2e) were either represented as electron densities decorated with their electrostatic potential (a) or as multi-class atom voxels with each voxel being assigned to the element types, for which at least one atom is closer than its van-der-Waals radius to a given voxel (b). Regardless of the representation, an additional channel representing an implicit solvent, as well as 12 or 24 *hidden-channels* and an *alive-channel* were appended.

## 2   Related Work

### 2.1   Cellular Automata and Neural Cellular Automata

Cellular Automata (CA), as originally introduced by Von Neumann and Burks [19] were used to study the developmental processes of multicellular organisms. Classically represented on a 2D grid, cells (as individual grid points) are periodically updated in parallel by a predefined set of update rules. These update rules usually consider the direct environment of a cell to update a cell's state in discrete time steps [4]. The set of states a cell can take is usually as simple as a binary representation of dead (0) and alive (1). In recent years, with the surge in developments of machine learning algorithms, architectures, and dedicated hardware, extensions of Cellular Automata with deep neural networks have been established. The simple update rules acting on the states of cells have been replaced with a deep neural network, in which the update rules can either be fixed or learned through established techniques, such as gradient descent-based optimization. Instead of a binary state each pixel (in 2D) or voxel (in 3D) is represented by a vector. A *alive-channel* guides the update behaviour of a cell's state and *hidden-channels* act as a cell's memory without explicit meaning [9]. The target channels (corresponding to the binary state in classical CAs) could be RGB-values in case of image generation, or a one-hot-encoding for discrete object generation tasks (Figure 1). In a previous work Sudhakaran et al. [17] proposed an extension of 2D NCAs [9] to 3D object generation to construct commonly found structures in Minecraft. They showed that NCAs are capable of accurately generating complex structures consisting of up to 3584 blocks with up to 50 unique block types, and Najarro et al. [10] implemented NCAs as Hypernetworks to solve reinforcement learning tasks. Several further studies have shown that NCAs are capable of generating a diverse set of 2D images [12], steering the behaviour of 2D objects [13], and generating video sequences [11].

### 2.2   Relation to Deep Learning

A wide variety of deep learning methods have recently been developed for numerous molecule representations. Transformers, as employed in text generation, have been applied to textual representations such as SMILES, SMARTS or even entire amino acids chains [16, 21, 3]. Graph Neural Networks (GNN) have been applied to graph representations of molecules [15, 14, 22], and Convolutional Neural Networks (CNN) to pixel and voxel representations [23, 6, 7]. The neural architecture of

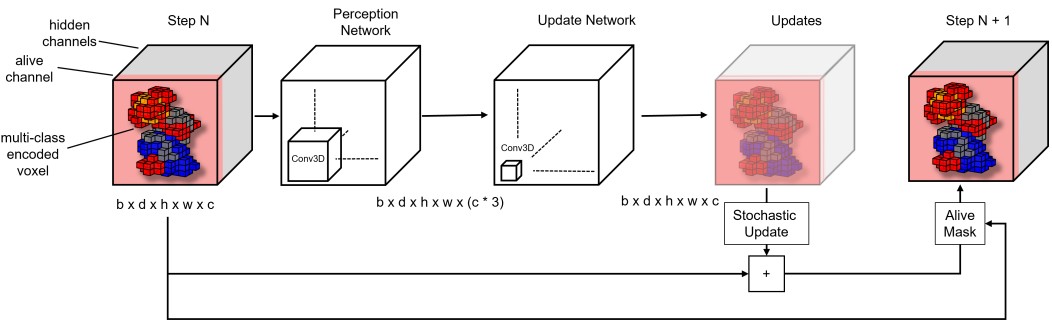

Figure 2: Architecture of the 3D-Neural Cellular Automata adapted from Sudhakaran et al. [17]

the NCAs implemented in this work is strongly connected to CNNs. Instead of updating the model weights, after a single pass over a minibatch, the exact same weight matrices and kernels are applied 10th or even 100th of times to morph an input into a target structure. Because of the close connection to CNNs, Mordvintsev et al. [9] termed NCAs in Deep Learning terms as *Recurrent Neural Networks with per-pixel Dropout*.

## 3  Methods

### 3.1  Data Representation

All molecules were either represented as voxelized electron densities decorated with their electrostatic potential or as multi-class atom voxels representing element types which are closer than their van-der-Waals radius to the centre of a voxel (Figure 1). Electron densities were calculated using the GFN2-xTB method implemented in xtb [1], and electrostatic potentials were calculated with ORBKIT [5]. The cut-off value for electron densities was set to 0.0001 au. Voxels with smaller values were set to 0. The voxelized electron densities were then decorated with electrostatic potentials by taking their Hadamard product. The decorated electron densities were split into two channels, representing voxels with negative and positive electrostatic potential. An additional channel was added which may be interpreted as a *solvent-channel*. The *solvent-channel* was set to 1 for voxels that were not occupied by electron densities. An additional channel indicates whether a voxel is dead or alive (*alive-channel*). Initially, alive voxels are those that are not solvent or have at least one non-solvent voxel as their direct neighbour. 12 or 24 *hidden-channels* (see Table 1 Appendix) were appended to each voxel's vector, and initialized to 0, if the voxel is dead, and 1 otherwise. For multi-class atom voxel representations, each voxel was represented with a channel for each element type in the considered system. The value of each channel with at least one atom of a given element type which is closer than its van-der-Waals radius to the centre of that voxel was set to 1, and 0 otherwise. Additional *solvent-*, *alive-*, and *hidden-channels* were appended as described above.

### 3.2  Model Architecture

The model architecture was closely adapted from Sudhakaran et al. [17] and is shown in Figure 2. In summary, the Perception- and Update-Networks were implemented as 3D convolutional layers. The *kernel-size*, *stride*, and number of *output-channels* were 3, 1, and 3 * #cell-state-channels for the Perception-Network, respectively. The Update-Network consisted of two 3D convolutional layers with *kernel-size* 1, and *stride* 1. The cell states were stochastically updated by setting half of the updates to zero before adding them to the input representation. Finally, an *alive mask* is applied ensuring that updates are only applied to cells that have at least one alive neighbouring cell.

### 3.3  Training

Training of the NCAs was performed via supervised learning. Depending on the task a target entity was either constructed from a single living cell, a molecule with missing parts, an interaction partner, or an isomer. The loss was calculated as the sum of an Intersection over Union (IOU) as defined

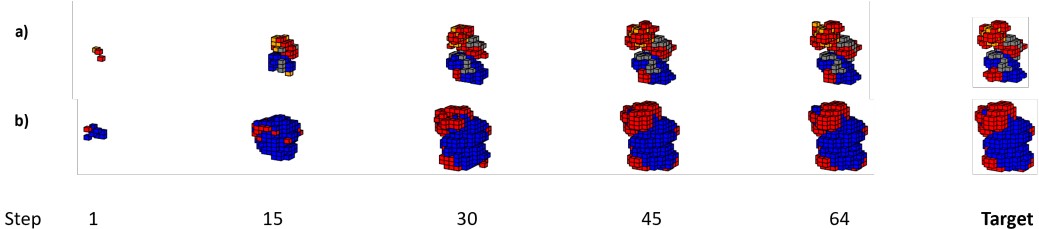

Figure 3: Regrowing molecules from a single seed block. Similar to Mordvintsev et al. [9] and Sudhakaran et al. [17], the *hidden-channels* and *alive-channel* of a single voxel in the centre of the grid were set to 1. An NCA was then trained to reconstruct the target structure from the seed voxel represented as a) multiclass-atom voxels or b) voxelized electron densities decorated with electrostatic potential.

in [17] and the Mean Squared Error (MSE) between cells non-solvent, visual channels and the corresponding target channels. The gradients were accumulated for 48 to 64 forward passes before updating the model. Unlike Sudhakaran et al. [17] and Mordvintsev et al. [9] we did not implement a training pool.

### 3.4 Experimental Details

All models were either trained on a single NVIDIA GeForce RTX 2080 Ti or NVIDIA TITAN RTX graphics card. Experiments were run for 5k to 20k epochs. System and model details are summarized in Table 1 and Table 2 (see Appendix).

## 4 Results

### 4.1 Growing voxelized Molecule Representations

To Verify that NCAs can generate accurate 3-dimensional molecule representations, the first task was to generate guanosine diphosphate (GDP; structure extracted from PDB 1d2e) from a single seed voxel. GDP was voxelized into electron densities decorated with their electrostatic potential and multi-class atom representations as described above. The grid resolution was set to 1 Å. Calculation of the electron density and electrostatic potential was initially performed with a resolution of 0.5 Å. A MaxPool3D operation as implemented in PyTorch with a *kernel_size* of $2x2x2$ was then applied to arrive at the desired resolution. To initiate the reconstruction of the voxelized targets, a single seed block was placed in the centre of the grid, by setting its *hidden-channel* values, and the *alive-channel* to 1, while setting every other channel and voxel to 0. The NCA was then trained to reconstruct the voxelized molecule representations. As shown in Figure 3, the NCA learned to accurately reconstruct either representation of GDP. Similar to what has been observed previously, the training occasionally suffers from sharp increases in loss (compare Appendix A1) [9, 17]. Even though the training of the NCAs was unstable at times, the loss quickly converged back again to accurately reconstruct the molecular representations, confirming that NCAs could be used to work on molecular problem settings.

### 4.2 Protein Pocket conditioned Molecule Generation

In many biological signal-processing pathways, and drug development tasks, interactions between proteins and small-molecules are of central importance. A recent trend in target-based drug development has thus focused on generating *de-novo* inhibitors, where the molecular generation process is conditioned on a given biological target. In analogy to that we set out to train a NCA, not to generate a molecule representation from a single seed block in isolation, but instead starting from a protein pocket. To do so, a 6 Å region around the centre of mass of GDP in the mitochondrial elongation factor thermo unstable (EF-Tu; PDB 1d2e) was extracted. The extracted complex was then voxelized using the multi-class atom voxel schema. To reconstruct the small-molecule interactor,

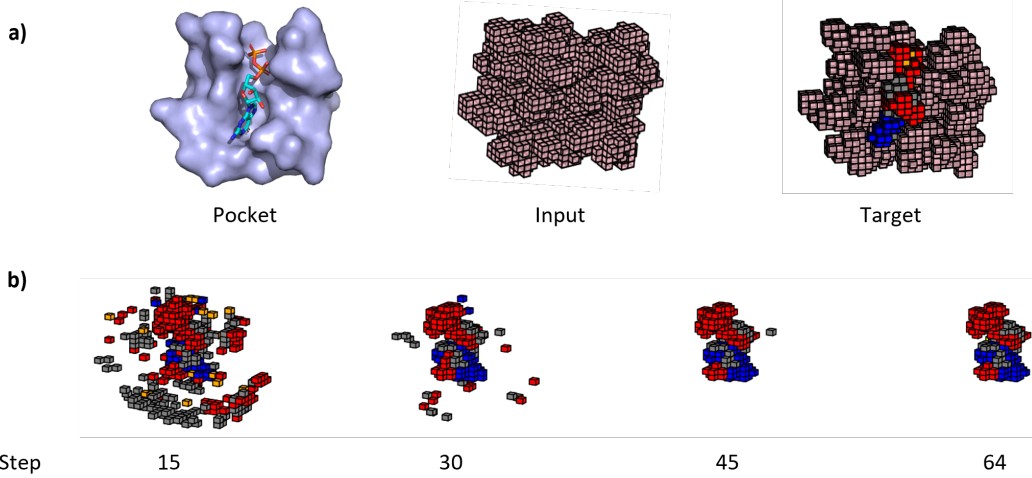

Figure 4: Protein pocket conditioned generation of molecules. a) The binding pocket of GDP in EF-Tu (PDB 1d2e) is shown on the left. The voxelized protein residues without GDP (middle) were passed into the NCA to reconstruct GDP within the binding pocket. The target structure of GDP in the binding pocket is shown on the right. b) Time series of GDP reconstruction. Voxels corresponding to protein residues were hidden for the sake of clarity.

all voxels occupied by GDP were set to 0. Other than in the reconstruction of GDP from a single seed block, no seed was placed in the centre of the box. Instead, the voxelized binding pocket may be considered as the seed. The target of the NCA was thus to reconstruct GDP in the binding pocket while keeping the voxels corresponding to the binding pocket static. Two additional loss terms were included to only consider the voxels that are not occupied by the binding pocket. After masking the voxels of the binding pocket, the MSE and IOU of the remaining voxels were calculated. As shown in Figure 4, the NCA was able to almost completely reconstruct the GDP in the binding pocket. The voxels corresponding to the binding pocket stayed mostly alive. While the training to regenerate GDP from a single seed showed only occasional spiked in the loss, the reconstruction in the binding pocket of EF-TU, proved to be rather unstable (Appendix A2). Even though the NCA was not able to converge to find a stable set of parameters, low-loss checkpoints of the model showed that the NCA learned to reconstruct GDP in the binding pocket. Notably, the NCA struggled to reconstruct voxels corresponding to phosphorus atoms which was the only atom type not present in the protein pocket, but only in GDP.

### 4.3 Protein Backbone Reconstruction

We also tested the NCAs capabilities of reconstructing missing parts of biomolecules on the example of missing residues of protein backbones. While many techniques have been established to reconstruct or model missing residues for proteins [2, 20], we argue that reconstructing missing parts of a protein backbone serves as a prime example to test the generalizability of the learned update rules of NCAs because of the structural similarities between proteins. To do so, we choose the structures of three proteins (PDB: 1aho, 1sps1, and 3nir) and deleted 4 amino acids from structurally diverse regions (beta-sheet, alpha-helix, and loop-region; compare Figure 5a). Before voxelizing, all sidechain atoms were removed. The protein backbones were then voxelized into multi-class atom representations. Unlike in previous studies, where typically a single NCA is trained to generate a single 3D object, a single NCA was trained to reconstruct the missing regions of the three proteins simultaneously. As shown in Figure 5c, the NCA learned to almost perfectly reconstruct the missing regions of the proteins, while keeping the rest of the protein backbone static, showing that NCAs have the capacity to learn universal rules on voxelized inputs.

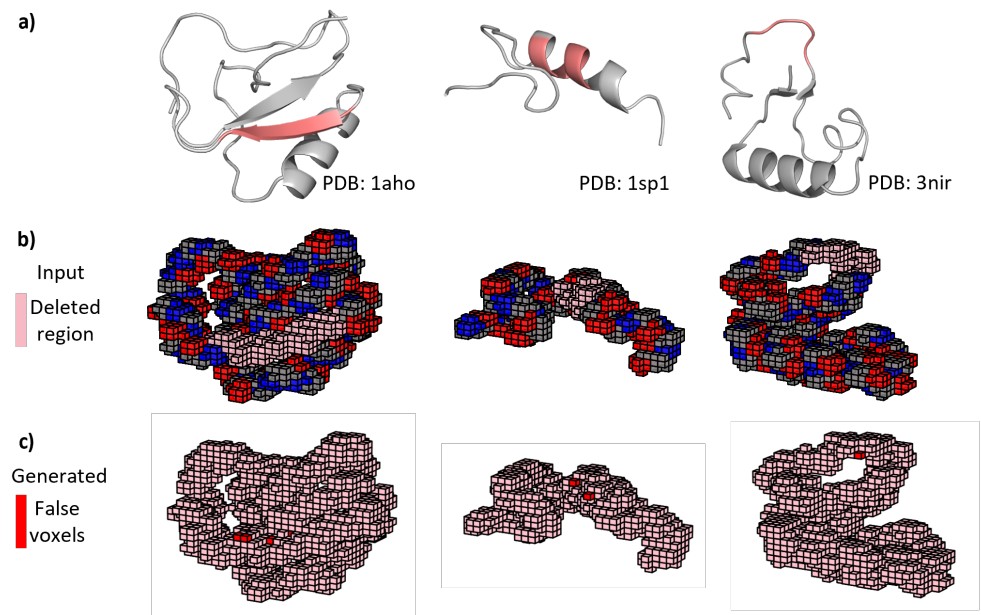

Figure 5: Reconstruction of protein backbones. The backbone atoms of residues 33-37 (PDB 1aho), residues 18-22 (PDB 1sp1), and 35-38 (PDB 3nir) highlighted pink in a) and b) were removed. A single NCA was then trained to simultaneously reconstruct the missing residues of all three proteins. c) Reconstruction of the protein backbone by the NCA. The missing residues were almost perfectly reconstructed. Falsely generated voxels are shown in red, and correct voxels in pink

## 4.4 Goal-guided Transformations

In [18] it was shown, that NCAs can be trained to follow goal-guided behaviour. They showed that different 2D structures could be morphed into each other by applying learned perturbation, to the *hidden-channels* of an input object. Inspired by that we adjusted our NCA to model a light-induced cis-trans isomerisation. Similar to what has been done by Sudhakaran et al. [18] we used a perturbation which was applied to the *hidden-channels* of all alive voxels to steer the behaviour of the system. Unlike described above we did not use learned perturbations but instead chose sine and cosine encodings to model a light switch (compare Appendix Figure A5). We chose the Stilbene derivative 1,1'-[($E$)-1,2-Ethenediyl]bis(3,5-dimethylbenzene) to model the isomerisation. Cis- and trans-isomers were generated using the conformer generation functionality implemented in RDKit [8]. Electron densities decorated with electrostatic potential were then generated as described above for both isomers. To model the physical transformations induced by turning the light switch on and off, the model should learn to morph the trans-isomer into the cis-isomer and simultaneously keep a given cis-isomer in a stable conformation. Likewise, a cis-isomer should be morphed into a trans-isomer and a trans- isomer should be in a stable conformation if the perturbation corresponding to the light switch being switched off is applied.

While simultaneously training the NCA to morph the trans-isomer into the cis-isomer and keep the cis-isomer stable (light switch on) or vice versa (light switch off), the model showed difficulty in learning the desired behaviour. We thus chose to first pretrain the NCA to morph trans to cis and cis to trans upon application of the given perturbation, and then fine-tune on all tasks simultaneously. Following this strategy, the model learned to reach the desired behaviour within 2500 pre-train epochs, and 2500 fine-tune epochs (compare Figure 6 and Appendix A4 for the loss).

## 5 Discussion and future Work

We have shown that the recent developments in NCAs for 3D object generation can be applied to a diverse set of molecular problems using simple voxelized multi-class atom representations and

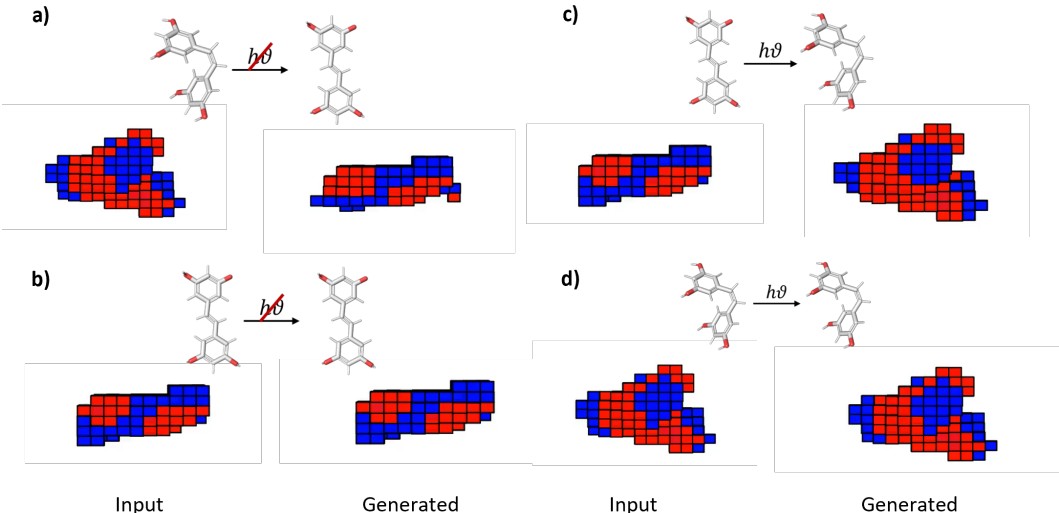

Figure 6: Light induced cis-trans isomerisation of 1,1'-[(*E*)-1,2-Ethenediyl]bis(3,5-dimethylbenzene). The fine-tuned NCA learned to simultaneously transform the cis-isomer (a)) and keep the trans-isomer static (b)), when the embedding vector corresponding to the light switch being off was applied, and vice versa (c/d)).

computationally more expensive electron-density representations. Using these methods, we were able to extend existing NCA models to generate small-molecule interactors conditioned on protein pockets, reconstruct missing parts of protein backbones and model physical transformations. We were also able to show that NCAs can generalize beyond generating single structures in 3D. While this illustrates the wide range of applications for NCAs in molecular problem settings, it is important to note that the task we have presented may not yet reach the level of the current state-of-the-art in this particular domain. However, it does open doors to further exploration and improvement. One current limitation is the significant amount of computing required to train for even a few training examples as done in the reconstruction of protein backbones. We also experienced instabilities while training the NCA, in agreement with previous studies. To circumvent these instabilities Mordvintsev et al. [9], Sudhakaran et al. [17] implemented a sample pool of initially identical structures. The sample pool gets updated with the output of the NCA after each training step. The sample with the largest loss is then replaced with the initial structure to circumvent catastrophic forgetting. A similar strategy would likely lead to stabilization of the training in the presented work but remains to be tested in future works.

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

# Appendix

Table 1: Molecule Representation Details

| Task | No. Classes | Size (W x D x H) |
|---|---|---|
| Molecule from seed (multiclass atom voxels) | 5 | 13 x 13 x 20 |
| Molecule from seed (ED-ESP) | 3 | 13 x 13 x 20 |
| Molecule from protein | 5 | 30 x 30 x 30 |
| Protein backbone reconstruction | 4 | 40 x 40 x 40 |
| Cis-Trans Isomerisation | 3 | 20 x 20 x 13 |

Table 2: Experimental Details

| Task | No. hidden channels | Min & Max Steps | NCA Update Net layer 1 and 2 | Layer initialization stdev | Learning Rate | Batch size |
|---|---|---|---|---|---|---|
| Molecule from seed (multiclass- atom voxels) | 12 | 48, 64 | 42, 42 | 0.01 | 0.001 | 1 |
| Molecule from seed (ED-ESP) | 24 | 48, 64 | 42, 42 | 0.01 | 0.001 | 1 |
| Molecule from protein | 12 | 48, 64 | 42, 42 | 0.04 | 0.001 | 1 |
| Protein backbone reconstruction | 12 | 48, 64 | 34, 34 | 0.01 | 0.001 | 3 |
| Cis-Trans Isomerisation | 12 | 48, 64 | 56, 56 | 0.01 | 0.001 | 1 |

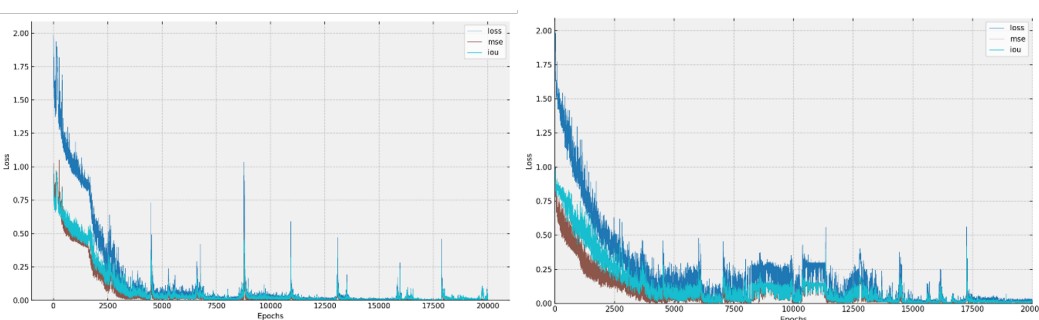

Fig. A1: Left: Loss of training NCA to regenerate electron density decorated with electrostatic potential of GDP from single seed voxel. Right: Loss of training NCA to regenerate multi-class atom voxel representation of GDP from single seed voxel.

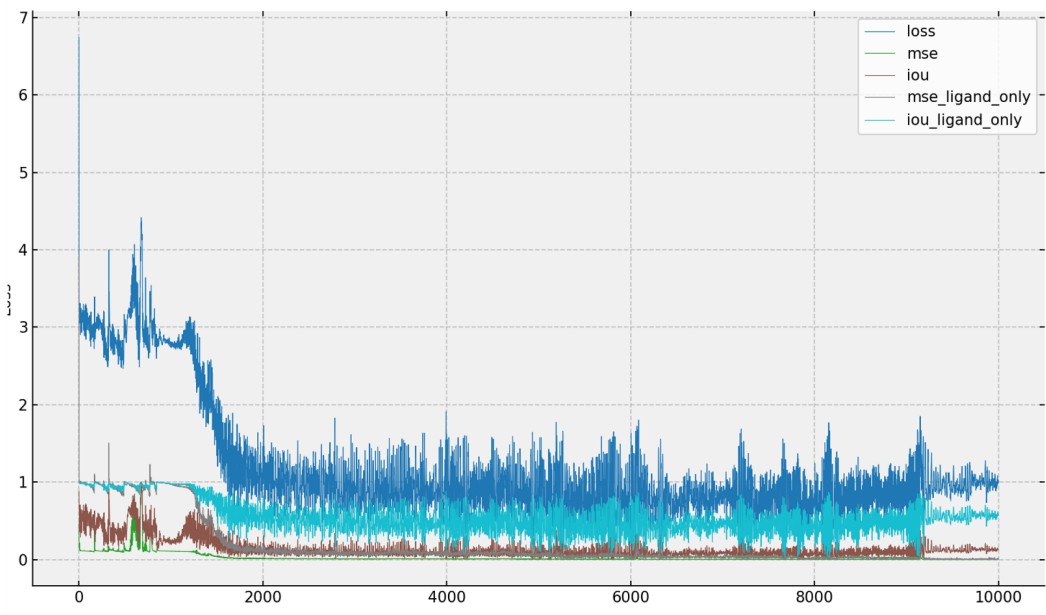

Fig. A2: Loss of training NCA to generate small-molecule interactor GDP from binding pocket of EF-Tu on multi-class atom voxel representation.

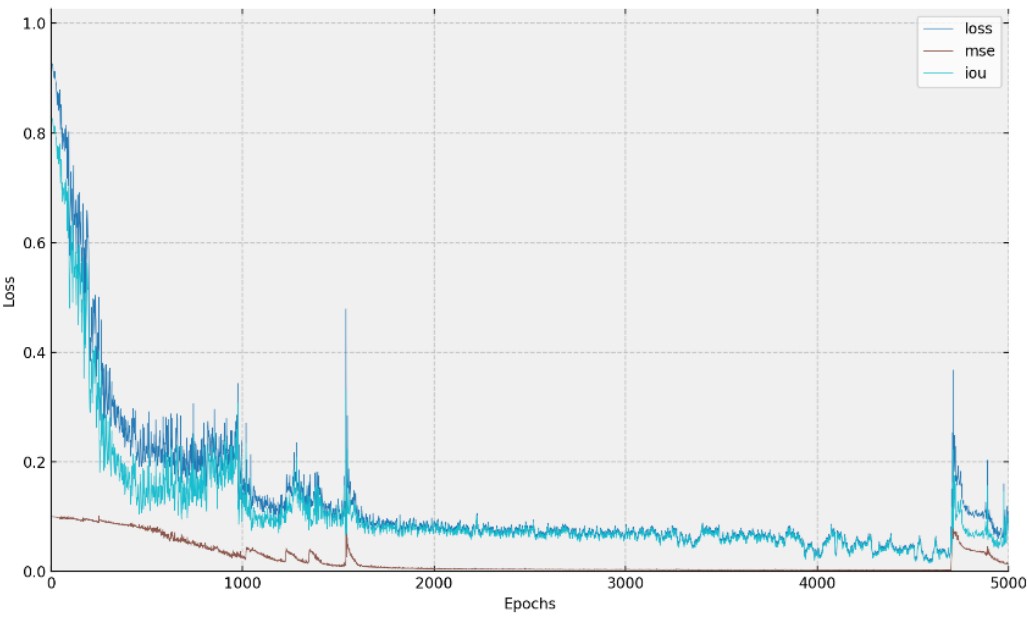

Fig. A3: Loss of training NCA to regenerate missing part of protein backbones on multi-class atom voxel representation.

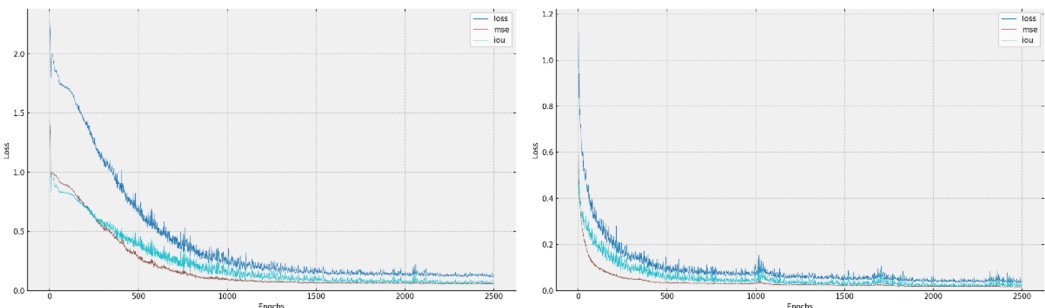

Fig. A4: Left: Loss of pretraining NCA to transform cis- to trans-isomer and vice-versa after application of embeddings. Right: Loss of fine-tuning NCA to additionally keep cis- or trans-isomer stable, when the respective embeddings are being applied.

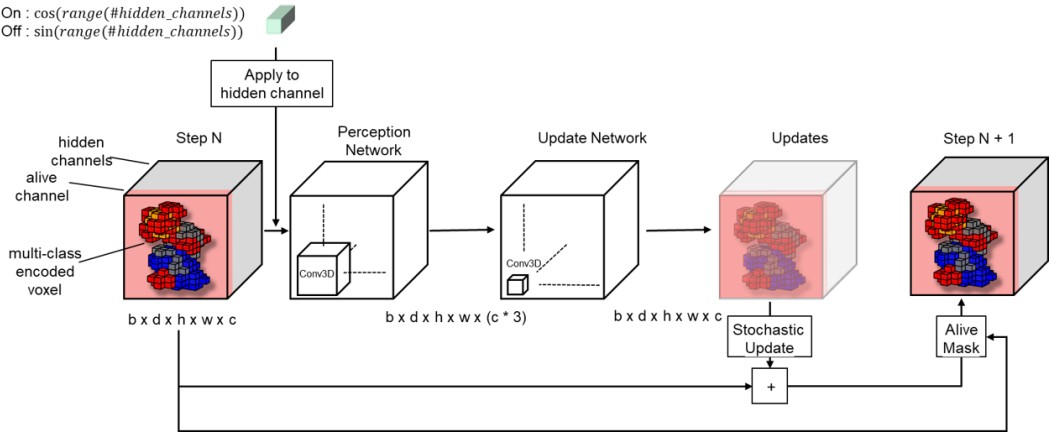

Fig. A5: Adapted model architecture for application of external forces. Similar to [18] an embedding vector was applied to all hidden channels of the input voxels by element-wise multiplication.

