# OpenReview forum: "Exploring the applications of Neural Cellular Automata in molecular sciences"
_NeurIPS.cc/2023/Workshop/AI4Science — NeurIPS2023-AI4Science Poster_

### Official Review · Reviewer_1aQ9 · 2023-10-25
**Comprehensive and well-written**

**Rating:** 7
**Confidence:** 1

**Review:**

Nice application and adaptation of Cellular Automata

---

### Meta-Review · Area_Chair_CvnA · 2023-10-26

**Recommendation:** Accept (Poster)
**Confidence:** 3

**Metareview:**

The paper explores a novel application of Neural Cellurar Automata (NCA) to tasks on molecules. Interestingly, NCA are capable of generating quite complex signals from images to 3d structures. The Authors demonstrate that NCA can be useful for generating various signals representing molecular structures (the electron density, ligand in a binding pocket, and others). One area for improvement is better demonstrating or arguing for what are unique advantages of using this approach vs others (e.g. diffusion). All in all, it is a solid contribution and I am happy to recommend acceptance of the paper.